# Comparative Study on Toughening Effect of PTS and PTK in Various Epoxy Resins

**DOI:** 10.3390/polym13040518

**Published:** 2021-02-09

**Authors:** Woong Kwon, Minwoo Han, Jongwon Kim, Euigyung Jeong

**Affiliations:** 1Department of Textile System Engineering, Kyungpook National University, Daegu 41566, Korea; kwoong7242@naver.com (W.K.); gksalsdn93@gmail.com (M.H.); 2Department of Fiber System Engineering, Yeungnam University, Gyeongsan 38541, Korea

**Keywords:** epoxy resin, toughening agent, fracture toughness, polytriazoleketone, polytriazolesulfone

## Abstract

This study investigated the toughening effect of in situ polytriazoleketone (PTK) and polytriazolesulfone (PTS) toughening agent when applied to various epoxy resins, such as diglycidyl ether of bisphenol A (DGEBA), diglycidyl ether of bisphenol F (DGEBF), and triglycidyl *p*-aminophenol (TGAP) with 3,3′-diaminodiphenylsulfone as a curing agent. The fracture toughness, tensile properties, and thermal properties of the prepared epoxy samples were evaluated and compared. When PTK was mixed with DGEBF, the fracture toughness was improved by 27% with 8.6% increased tensile strength compared to the untoughened DGEBF. When PTS was mixed with TGAP, the fracture toughness was improved by 51% without decreasing tensile properties compared to the untoughened TGAP. However, when PTK or PTS was mixed with other epoxy resins, the fracture toughness decreased or improved with decreasing tensile properties. This is attributed to the poor miscibility between the solid-state monomer of PTK (4,4′-bis(propynyloxy)benzophenone (PBP)) or PTS (4,4′-sulfonylbis(propynyloxy)benzene (SPB)) and the epoxy resin, resulting in the polymerization of low molecular weight PTK or PTS in epoxy resin. Therefore, the toughening effect of PTK or PTS can be maximized by the appropriate selection of epoxy resin based on the miscibility between PBP or SPB and the resin.

## 1. Introduction

Epoxy resins are used as matrices in fiber-reinforced composite materials because of its excellent thermal, mechanical, and chemical properties. Among the various epoxy resins, highly crosslinked epoxy resins are widely used in aerospace application [1,2,3,4]. However, highly crosslinked epoxy resins are brittle materials with low fracture toughness, restricting its structural engineering applications in aerospace industries [5]. Therefore, many studies have attempted to improve the fracture toughness of highly crosslinked epoxy resins using various toughening agents.

To improve the fracture toughness of epoxy resins, the toughening agents are used with inorganic particles [6,7,8,9,10], rubbers [11,12,13,14,15], and engineering thermoplastics [16,17,18]. Inorganic particles such as silica, carbon nanotube, and graphene improve the fracture toughness, thermal stability, and elastic modulus of epoxy resin. However, uniform dispersion of inorganic particles in the epoxy resin is difficult to achieve [6,7,8,9,10]. Rubbers such as carboxy-terminated butadiene-acrylonitrile and amine-terminated butadiene-acrylonitrile improve fracture toughness. However, rubbers decrease the thermal properties of epoxy resins due to their lower glass transition temperatures (T_g_) [11,12,13,14,15]. Engineering thermoplastics such as poly(ethersulfone) (PES) and poly(etheretherketone) improve the fracture toughness without decreasing the thermal and mechanical properties. However, when engineering thermoplastics are applied to the epoxy resin as a toughening agent, the viscosity of the epoxy resin drastically increases, which causes problems in the molding process [16,17,18].

Recently, in situ-type thermoplastic toughening agents, polytriazoleketone (PTK) and polytriazolesulfone (PTS) polymer, were developed to improve the fracture toughness without increasing the viscosity of the epoxy resin [19,20,21,22,23,24]. These are added to the monomer forms in the epoxy resin and are polymerized via azide-alkyne click reaction during epoxy curing reaction. Ying et al. reported the toughening effect of in situ PTS-type toughening agents in triglycidyl *p*-aminophenol (TGAP) with 4,4′-diaminodiphenylsulfone (DDS) curing agent. They found that the fracture toughness of TGAP increased by 100% compared to that of the untoughened epoxy resin [19]. Lee et al. reported the toughening effect of in situ PTK-type toughening agents. When PTK was mixed with TGAP/4,4′-DDS, the fracture toughness of TGAP increased by 91% compared to that of the untoughened epoxy resin [20]. When comparing the toughening effect of PTK and PTS with PES, which is commercial engineering plastic toughening agents, PTK or PTS showed better toughening effect than PES, and toughening effects of in situ PTK- or PTS- type toughening agents were different [19,20]. In addition, Lee et al. reported the toughening effect of in situ PTS-type toughening agents according to the type of epoxy resin with 4,4′-DDS curing agent. Applying the toughening agents to diglycidyl ether of bisphenol A (DGEBA) and TGAP, the fracture toughness of DGEBA and TGAP improved by 25% and 67%, respectively. However, when PTS was mixed with diglycidyl ether of bisphenol F (DGEBF), the fracture toughness of DGEBF was decreased by 7% [21]. The toughening effect of in situ-type toughening agent differs depending on the miscibility between the epoxy resin and the solid-state monomer of in situ-type toughening agent. Therefore, PTK may exhibit a higher toughening effect than PTS depending on the type of epoxy resin. To use PTK as a toughening agent to the epoxy resin, it is necessary to optimize its toughening effect depending on the type of the epoxy resin. In addition, the toughening effect of PTS-toughened DGEBA or PTS-toughened DGEBA, DGEBF, and TGAP were studied using 4,4′-DDS as a curing agent. When using 3,3′-DDS curing agent, it has better fracture toughness than 4,4′-DDS curing agent [22]. However, when 3,3′-DDS is used as a curing agent, the toughening effect of PTK or PTS according to the type of epoxy resin has not been reported.

This study aims to compare the toughening effect of PTK or PTS according to the type of epoxy resin with 3,3′-DDS. PTK or PTS was applied to three types of epoxy resin (DGEBA, DGEBF, and TGAP) with 3,3′-DDS as a curing agent. The fracture toughness of epoxy resins applied with PTK or PTS was evaluated and compared. Furthermore, the effect of PTK or PTS on the tensile and thermal properties of the epoxy resin was evaluated. The effect of the toughening agents on the fracture toughness, tensile strength, and tensile modulus were different for each type of epoxy resin. To analyze the difference in fracture toughness, tensile strength, and tensile modulus, the solubility parameter was calculated. The calculated miscibility by the solubility parameter between the epoxy resin and solid-state monomer of the PTK or PTS was compared and the toughening effect was investigated to optimize the toughening effect depending on the type of epoxy resin.

## 2. Materials and Methods

### 2.1. Materials

Diglycidyl ether of bisphenol A (DGEBA) and diglycidyl ether of bisphenol F (DGEBF) were purchased from Kukdo Chemical Co. Ltd. (Seoul, Korea). Triglycidyl *p*-aminophenol (TGAP) was purchased from Kukdo Finechem Co. Ltd. (Busan, Korea) and 3,3′-diaminodiphenylsulfone (3,3′-DDS) was purchased from Wakayama Seika Kyoto Co. Ltd. (Kyoto, Japan). Sodium azide (99.5%), α, α’-dichloro-*p*-xylene (98%), propargyl bromide (80 wt.% in toluene), 4,4′-dihydroxybenzophenone (99%), and 4,4′-sulfonyldiphenol (99%) were purchased from Sigma-Aldrich (St. Louis, MO, USA). Acetonitrile (95%), dichloromethane (99%), hexane (95%), and acetone (95%) were purchased from Duksan Chemical Co. Ltd. (Ansan, Korea).

### 2.2. Synthesis of the In Situ Polytriazoleketone and Polytriazolesulfone Toughening Agents

#### 2.2.1. Synthesis of the 1,4′-Bis(azidomethyl)benzene (*p*-BAB)

20.0 g of α, α’-dichloro-*p*-xylene, 29.7 g of sodium azide, and 700 mL of acetonitrile were added into a three-necked round flask and stirred at 78 °C for 48 h. The mixture was filtered and was then extracted with dichloromethane and water using a separating funnel to remove the unreacted sodium azide. The solvent was removed and the residue was purified using column chromatography (eluent: dichloromethane/n-hexane = 7/3). Finally, after evaporating, the diazide monomer was obtained as a liquid.

#### 2.2.2. Synthesis of the 4,4′-Bis(propynyloxy)benzophenone (PBP)

10.0 g of 4,4′-dihydroxybenzophenone, 31.8 g of potassium carbonate, and 160 mL of acetone were added into a three-necked round flask. 27.4 mL of propargyl bromide was then added dropwise and the mixture was stirred at 80 °C for 48 h under a nitrogen atmosphere. The mixture was filtered and precipitated in methanol. Finally, the PBP, which is a dialkyne monomer, was obtained as a white solid.

#### 2.2.3. Synthesis of the 4,4′-Sulfonylbis(propynyloxy)benzene (SPB)

20.0 g of 4,4′-sulfonyldiphenol, 54.6 g of potassium carbonate, and 700 mL of acetone were added into a three-necked round flask. 35.2 mL of propargyl bromide was then added dropwise and the mixture was stirred at 80 °C for 48 h under a nitrogen atmosphere. The mixture was filtered and precipitated in methanol. Finally, the SPB, which is a dialkyne monomer, was obtained as a white solid.

#### 2.2.4. Synthesis of the PTK and PTS

To confirm the chemical structure and determine the glass transition temperature and molecular weight of PTK or PTS, *p*-BAB and PBP (or SPB) were polymerized in dimethylsulfoxide (DMSO). The temperature of reaction is similar to the epoxy curing condition. To synthesize PTK, 0.44 g of PBP (1.53 × 10^−3^ mol) and 0.29 g of *p*-BAB (1.53 × 10^−3^ mol) were dissolved in 10.0 mL of DMSO in a two-necked flask. The reaction mixture was stirred and heated to 180 °C for 3 h. After the reaction, the mixture was precipitated with tetrahydrofuran (THF) and filtered. The powder was then washed with THF and dried at 70 °C overnight. Finally, PTK was obtained as a brown solid. To synthesize PTS, 0.50 g of SPB (1.53 × 10^−3^ mol) and 0.29 g of *p*-BAB (1.53 × 10^−3^ mol) were dissolved in 10.0 mL of DMSO in a two-necked flask. The reaction mixture was stirred and heated to 180 °C for 3 h. After the reaction, the mixture was precipitated with THF and filtered. The powder was washed with THF and dried at 70 °C overnight. Finally, PTS was obtained as a brown solid. Figure 1 illustrates the reaction scheme of PTK and PTS.

### 2.3. Preparation of the Epoxy Samples

The molar equivalent ratio of the epoxy resin and curing agent was adjusted to 1: 0.85. 26.8, 33.0, and 55.8 phr of 3,3′-DDS were mixed with DGEBA, DGEBF, and TGAP, respectively. 0.5 phr of *p*-BAB/PBP (or SPB) was added to the mixture of epoxy resin and curing agent. The mixture was stirred with a mechanical stirrer at 90 °C for 1 h and then degassed under reduced pressure. After, the mixture was cured at 180 °C for 1 h and 210 °C for 2 h to prepare the PTK- or PTS-toughened epoxy. The untoughened epoxy samples were labeled as DGEBA, DGEBF, and TGAP. The PTK-toughened epoxy samples were labeled as DGEBA/PTK, DGEBF/PTK, and TGAP/PTK, and the PTS-toughened epoxy samples were labeled as DGEBA/PTS, DGEBF/PTS, and TGAP/PTS.

### 2.4. Characterization and Evaluation of the PTK, PTS, and Prepared Epoxy Samples

The FT-IR spectra of *p*-BAB, PBP, SPB, PTK, PTS, and the prepared epoxy samples were obtained using an FT-IR spectrophotometer (Nicolet iS5, Thermo Fisher Scientific, Waltham, MA, USA) equipped with an attenuated total reflection accessory (iD7 ATR, Thermo Fisher Scientific, Waltham, MA, USA). The ^1^H-NMR spectra of *p*-BAB, PBP, and SPB were obtained using a NMR spectrometer (AVANCE III HD 400, Bruker, Rheinstetten, Germany) and DMSO-*d*_6_ was used as a solvent. The glass transition temperatures (T_g_s) of PTK and PTS were investigated using differential scanning calorimetry (DSC, Q2000, TA instruments, New Castle, DE, USA) from 25 to 300 °C with a heating rate of 5 °C/min under nitrogen atmosphere. The molecular weights of PTK and PTS were obtained through gel permeation chromatography (GPC, YL9100, Youngin Instrument, Anyang, Korea) equipped with a refractive index (RI) detector (YL9170, Youngin Instrument, Anyang, Korea). The fracture surfaces of the prepared epoxy samples were investigated with a field emission-scanning electron microscopy instrument (FE-SEM, S-4800, Hitachi, Tokyo, Japan) with an accelerating voltage of 3 kV. The tensile properties of the prepared epoxy samples were evaluated on the ASTM D638 with a crosshead speed of 2 mm/min using a universal test machine (UTM, AG-5kNX, Shimadzu, Kyoto, Japan). The fracture toughness of the prepared epoxy samples was evaluated on the ASTM D5045 with a crosshead speed of 1 mm/min using SENB (single end notched bending) specimen and UTM. The T_g_s of the prepared epoxy samples were investigated using dynamic mechanical analysis (DMA, Discovery DMA 850, TA instruments, New Castle, DE, USA) from 25 to 300 °C with a heating rate of 2 °C/min and a frequency of 1 Hz under nitrogen atmosphere. The mixtures of epoxy resin and toughening agent before curing were observed with an optical microscope (SMZ100, Nikon Co., Tokyo, Japan).

## 3. Results and Discussion

### 3.1. Chemical Structure and Reaction Confirmation of PTK and PTS

Figure 2 illustrates the reaction schemes of *p*-BAB, PBP, and SPB. Figure 3, Figure 4 and Figure 5 show the FT-IR and ^1^H-NMR spectra of *p*-BAB, PBP, and SPB, respectively.

In Figure 3, the peak at 717 cm^−1^ arising from the chloromethyl group of 1,4-bis(chloromethyl) benzene is not observed after the reaction. New peaks appeared at 1250 and 2100 cm^−1^ after the reaction and were attributed to the C–N and –N_3_ from the azide group of *p*-BAB, respectively. In the ^1^H-NMR spectrum of *p*-BAB, the signals are observed at δ_ppm_ = 4.5 from methylene proton and 7.4 from aromatic proton. The integral ratio of each peak is 4(A):4(B), which are in excellent agreement with the theoretical ratio of each peak.

In Figure 4, the peak at 3380 cm^−1^ arising from –OH group of 4,4′-dihydroxybenzophenon is not observed after the reaction. New peaks at 2113 and 3262 cm^−1^ were attributed to the acetylene group of PBP. In the ^1^H-NMR spectrum, the signals are observed at δ_ppm_ = 3.3 from acetylene proton, 4.9 from methylene proton, 7.2 from aromatic proton next to the ether group, and 7.9 from aromatic proton next to the ketone group. The integral ratio of each peak is 2(A):4(B):4(C):4(D), which are in excellent agreement with the theoretical ratio of each peak.

In Figure 5, the peak at 3360 cm^−1^ arising from –OH group of 4,4′-sulfonyldiphenol is not observed after the reaction. New peaks appeared at 1237, 2078, and 3270 cm^−1^ after the reaction. The new peak at 1237 cm^−1^ was attributed to the C–O–C group of SPB, and the new peaks at 2078 and 3270 cm^−1^ were attributed to the acetylene group of SPB. In the ^1^H-NMR spectrum of SPB, the signals are observed at δ_ppm_ = 3.3 from an acetylene proton, 4.0 from methylene proton, 7.2 from aromatic proton next to the ether group, and 7.9 from the aromatic proton next to the sulfonyl group. The integral ratio of each peak is 2(A):4(B):4(C):4(D), which are in excellent agreement with the theoretical ratio of each peak.

### 3.2. Molecular Weight of PTK and PTS

The molecular weight of PTK or PTS, which was polymerized in DMSO, is analyzed using GPC. The polymerization temperature of PTK or PTS was similar to the epoxy curing reaction temperature. M_n_ of the polymerized PTK was 2.4 × 10^4^ g/mol, M_w_ was 3.4 × 10^4^ g/mol, and polydispersity (PDI) was 1.40. M_n_ of the polymerized PTS was 2.0 × 10^4^ g/mol, M_w_ was 2.6 × 10^4^ g/mol, and PDI was 1.32. M_w_ of PTK and PTS are higher than that of the commercial polyethersulfone (PES, M_w_ = 0.5 × 10^4^–1.5 × 10^4^ g/mol), which is an engineering thermoplastic toughening agent and is sometimes used as a copolymer with polyetherethersulfone (PEES) [19,20,25]. The molecular weight of PTK or PTS polymerized under DMSO was high enough to be used as a toughening agent. Furthermore, to obtain the high molecular weight of PTK or PTS via azide-alkyne click reaction, the molar equivalent ratio of the monomers should be 1:1, which is only achieved when the purity of the monomers is sufficiently high. The high molecular weight of PTK or PTS suggests that the monomers of PTK or PTS have enough purity.

### 3.3. Morphology of the Fracture Surfaces of the Prepared Epoxy Samples

The fracture surfaces of the prepared epoxy samples were observed by SEM to confirm the dispersion state of PTK or PTS in the epoxy resins, and the results are shown in Figure 6. The fracture surfaces of untoughened epoxy such as DGEBA, DGEBF, and TGAP were smooth and clean. In addition, the fracture surfaces of PTK- or PTS-toughened epoxy were also smooth and clean and PTK or PTS particles were not observed in the epoxy resin. PTK or PTS, which was polymerized via azide-alkyne click reaction during curing reaction in the epoxy resin, has been reported to be polymerized to a size of 50–200 nm [19]. Therefore, PTK or PTS was well-dispersed in a small size in the epoxy resin, which resulted in smooth and clean surface in SEM analysis.

### 3.4. Mechanical and Thermal Properties of the Prepared Epoxy Samples

Figure 7 presents the fracture toughness of the prepared epoxy samples. When PTK or PTS was mixed with DGEBA, fracture toughness of DGEBA/PTK and DGEBA/PTS was improved by 7%, compared to the untoughened DGEBA. When PTK or PTS mixed with DGEBF, fracture toughness of DGEBF/PTK was improved by 27%, whereas the fracture toughness of DGEBF/PTS was similar to that of the untoughened DGEBF. When PTK and PTS were mixed with TGAP, fracture toughness was improved by 58% and by 51%, respectively, compared to that of the untoughened TGAP. These results suggest that the toughening effect of PTK is equal to or better than that of PTS, and the toughening effect of PTK or PTS differs depending on the type of epoxy resin. In the previous study, when TGAP/3,3′-DDS was toughened with PES, which is commercial engineering thermoplastics, the fracture toughness improved by 7% [24]. Therefore, PTK or PTS shows sufficiently excellent toughening effect compared to PES.

The tensile properties of the prepared samples are shown in Table 1. The tensile strength of DGEBA/PTK (89.3 ± 6.4 MPa) was similar to that of the untoughened DGEBA (93.5 ± 13.7 MPa). The tensile strength of DGEBF/PTK (93.7 ± 15.0 MPa) was improved by 9%, whereas tensile strength of TGAP/PTK (100.9 ± 10.6 MPa) was decreased by 14% compared to the untoughened DGEBF (86.3 ± 3.4 MPa) and TGAP (116.7 ± 8.2 MPa), respectively. The tensile strength of DGEBA/PTS (94.1 ± 4.2 MPa) or TGAP/PTS (111.6 ± 11.9 MPa) was also similar to that of the untoughened epoxy resin (93.5 ± 13.7 MPa of DGEBA, 116.7 ± 8.2 MPa of TGAP), whereas DGEBF/PTS (94.1 ± 4.2 MPa) was improved by 13% compared to untoughened DGEBF (86.3 ± 3.4 MPa). The tensile moduli of the prepared epoxy samples increased by 4% to 11%, except for DGEBA/PTK and TGAP/PTK.

In order to PTK or PTS to have an excellent toughening effect without decreasing tensile properties, the monomer of PTK or PTS must be uniformly dispersed in the epoxy resin so that it can be polymerized with high molecular weight. When the monomer of PTK or PTS were not uniformly dispersed and resulted in low molecular weight PTK or PTS, the fracture toughness and tensile properties will be decreased or the fracture toughness will be increased with decreasing tensile properties [21]. When PTK was mixed with DGEBF or PTS was mixed with TGAP, excellent toughening effect was observed and the tensile properties were improved or maintained. This result suggested that a solid-state monomer of PTK (PBP) is uniformly dispersed in DGEBF, and a solid-state monomer of PTS (SPB) is uniformly dispersed in TGAP and polymerized to a high molecular weight during the epoxy curing reaction. On the other hand, when PTS was mixed with DGEBF, the fracture toughness was slightly decreased and tensile properties were improved. When PTK was mixed with TGAP, fracture toughness of TGAP was improved but the tensile properties were decreased. The molecular weight of PTK in TGAP was low, which resulted in increased fracture toughness of TGAP/PTK but the tensile properties decreased. This will be discussed in Section 3.5.

The thermal properties of PTK and PTS, which were polymerized in DMSO, were investigated using DSC and the results are shown in Figure 8. The T_g_ of PTK obtained by the reaction of *p*-BAB and PBP is 132.30 °C, while the T_g_ of PTS obtained by the reaction of *p*-BAB and SPB is 149.22 °C.

Figure 9 and Table 2 depict the dynamic mechanical properties of the prepared epoxy samples. The DGEBF epoxy sample with 3,3′-DDS curing agent showed the lowest T_g_ at 148.7 °C, while the TGAP epoxy sample showed the highest T_g_ at 215.5 °C. The PTK- and PTS-toughened epoxy resins exhibited 8–20 °C and 7–19 °C lower T_g_s compared to that of the untoughened epoxy because the T_g_ of PTK and PTS themselves were lower than that of untoughened epoxy. There was a slight decrease after toughening with PTK or PTS but excellent thermal properties were maintained. Except for DGEBF, DGEBA, TGAP/PTK, and TGAP/PTS, one peak was observed in the tan δ curves of the prepared epoxy samples. However, in the tan δ curves of DGEBF, DGEBA, TGAP/PTK, and TGAP/PTS, minor or shoulder peaks were observed. Minor or shoulder peak may appear due to the phase separation between the epoxy resin and the toughening agent or inhomogeneous crosslinking density [26,27,28]. In the morphological analysis (Figure 6), phase separation was not observed. The 3,3′-DDS used for curing the epoxy resin has two primary amine groups and it can react with four epoxide groups to form crosslinks. During the mixture process at 90 °C, the epoxide groups and hydroxyl groups of epoxy resins can react, increasing the molecular weight of epoxy resin itself. This will reduce the amount of epoxy groups and cause unreacted amine groups of the curing agent, resulting in the inhomogeneous crosslinking density. Therefore, minor peaks or shoulder peaks appeared due to the inhomogeneous crosslinking density [28].

### 3.5. Analyses of the Mechanical Properties of the Prepared Epoxy Samples

The in situ polymerization of *p*-BAB and PBP (or SPB) during epoxy curing reaction was confirmed using FT-IR to analyze the mechanical properties of the prepared epoxy samples. Figure 10 showed the acquired FT-IR spectra. The mixture of *p*-BAB, PBP (or SPB), epoxy resin, and curing agent is labeled as mixture and the cured mixture is labeled as specimen. As shown in Figure 10, the peak at 2100 cm^−1^ arising from the azide or acetylene of the *p*-BAB, PBP, and SBP disappeared after curing of epoxy resin. This result suggests that the *p*-BAB/PBP (or SPB) are successfully polymerized via azide-alkyne click reaction during epoxy curing reaction.

PTK and PTS polymerized in DMSO have a high molecular weight, and the monomers performed in situ click reaction during epoxy curing reaction (Figure 10). For PTK- or PTS-toughening agent, *p*-BAB as a liquid-state monomer and PBP (or SPB) as a solid-state monomer of PTK (or PTS) are added to the epoxy resin, and *p*-BAB and PBP (or SPB) are polymerized via azide-alkyne click reaction during epoxy curing reaction. However, the molecular weight of PTK or PTS polymerized in the epoxy resin was low, resulting in a toughening effect like plasticizers. The fracture toughness of epoxy resin can be improved with plasticizers, which are sometimes in monomer forms, resulting in reduced tensile properties [5]. To obtain a high molecular weight of the PTK or PTS polymerized via azide-alkyne click reaction of *p*-BAB and PBP (or SPB), a molar equivalent ratio of 1:1 should be obtained. Therefore, the molar equivalent ratio may be 1:1 when PBP (or SPB), which is a solid form of monomer, is sufficiently dissolved in the epoxy resin. On the other hand, when PBP (or SPB) is not sufficiently dissolved in the epoxy resin, each molecule of *p*-BAB and PBP (or SPB) is not well-mixed and the molar equivalent is not 1:1, resulting in a polymerized low molecular weight PTK or PTS. Therefore, the miscibility between the epoxy resin and the solid-state monomer of PTK or PTS is important.

To confirm the miscibility between PBP (or SPB) and the epoxy resins, the solubility parameters of PBP (or SPB), DGEBA, DGEBF, and TGAP were calculated by Fedors group contribution theory. Fedors group contribution theory is the simplest calculation method for solubility parameter and the related equation is shown in Equation (1) [29].
δ = (∑ΔE_coh_*/*V)^1/2^(1)
where δ is the solubility parameter, ΔE_coh_ is the cohesive energy, and V is the molar volume.

The solubility parameters of PBP, DGEBA, DGEBF, and TGAP calculated by Equation (1) were 23.2, 22.3, 23.4, and 24.3 (J/cm^3^)^1/2^, respectively. However, the solubility parameter of SPB was difficult to calculate with Fedors group contribution theory because ΔE_coh_ and V of the –SO_2_– group in SPB were not provided. Therefore, ΔE_coh_ and V of the SO_3_, SO_4_, and –SO_2_Cl were used instead of that of the –SO_2_– group. As a result, 24–25 (J/cm^3^)^1/2^ was obtained. In addition, SPB is best soluble in dimethylformamide (DMF), which has a solubility parameter of 24.7, so the obtained value of 24–25 (J/cm^3^)^1/2^ are reliable. As for the theoretical solubility parameter calculated by Fedors theory, the miscibility of PBP is best to DGEBF, while SPB is best to TGAP. Optical microscopy analysis was performed to confirm the compatibility predicted by the solubility parameter.

Figure 11 presents the optical microscope images of the mixture of epoxy and toughening agents to verify the miscibility calculated by solubility parameters. In the mixture of DGEBF/PBP, little particles were observed, whereas large particles were observed in DGEBA/PBP and TGAP/PBP. This result suggested that DGEBF showed the best miscibility with PBP and resulted in improved tensile properties and the most improved fracture toughness. On the other hand, in the mixture of TGAP/PBP, no particle was observed, whereas particles were observed in DGEBA/SPB and DGEBF/SPB. The miscibility of SPB is best with TGAP and resulted in the most improved fracture toughness among the samples.

## 4. Conclusions

The in situ PTK- and PTS-toughening agents were synthesized from *p*-BAB and PBP (or SPB). The PTK- and PTS-toughening agents were used in monomer forms and were polymerized by azide-alkyne click reaction during epoxy curing reaction. In various epoxy resins, the toughening effect of PTK was similar to or better than that of PTS. In addition, PTK or PTS showed different toughening effects depending on the type of epoxy resin. When PTK was mixed with DGEBF, the fracture toughness of DGEBF was improved by 27% with increased tensile properties of DGEBF. When PTS was mixed with TGAP, the fracture toughness of TGAP was improved by 51% with maintained tensile strength and increased tensile modulus. Except for PTK-toughened DGEBF and PTS-toughened TGAP, the fracture toughness decreased or improved with decreased tensile properties. This is attributed to the difference in miscibility between the solid-state monomer of PTK (PBP) or PTS (SPB) and the epoxy resin. When their miscibility is excellent, they are polymerized with a high molecular weight in the epoxy resin and the fracture toughness is improved without decreasing the tensile properties. On the other hand, when the miscibility is poor, they are polymerized with a low molecular weight and the fracture toughness is decreased or improved with decreasing tensile properties. Therefore, to analyze the miscibility, the solubility parameters of PBP, SPB, and various epoxy resin were calculated. The solubility parameter of PBP was closest to that of DGEBF, while the solubility parameter of SPB was closest to that of TGAP. When the solubility parameters of the solid-state monomer and the epoxy resin are close, high molecular weight PTK or PTS is polymerized. As a result, excellent toughening effect is shown without decreasing tensile properties. Therefore, when the solubility parameter between the epoxy resin and the solid-state monomer of the PTK or PTS is considered well, and in situ toughening agents are applied to the epoxy resin having excellent miscibility with toughening agent, excellent toughening effect can be obtained without decreasing tensile properties.

## Figures and Tables

**Figure 1 polymers-13-00518-f001:**
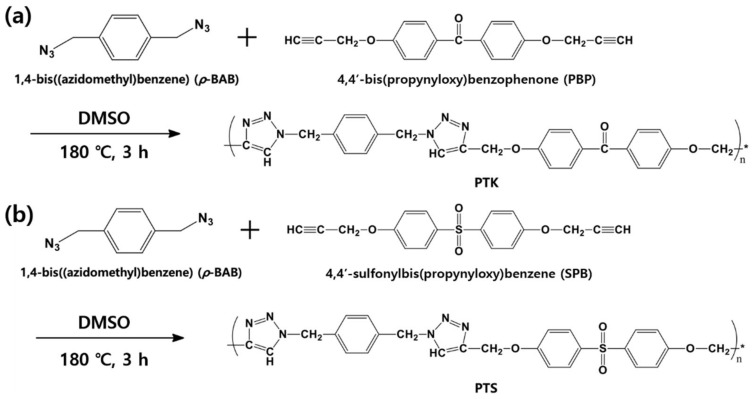
Reaction schemes of (**a**) PTK and (**b**) PTS.

**Figure 2 polymers-13-00518-f002:**
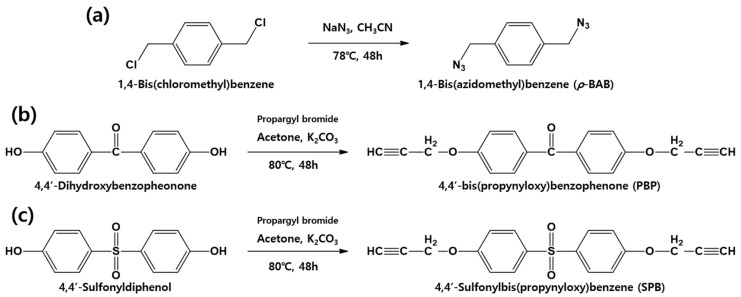
Reaction scheme of the *p*-BAB, PBP, and SPB: (**a**) *p*-BAB; (**b**) PBP; and (**c**) SPB.

**Figure 3 polymers-13-00518-f003:**
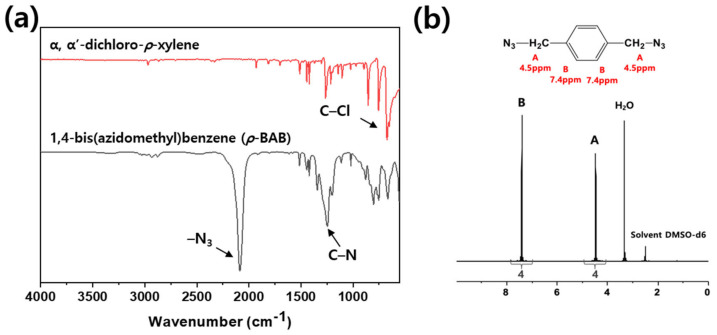
(**a**) FT-IR and (**b**) ^1^H-NMR spectrum of *p*-BAB.

**Figure 4 polymers-13-00518-f004:**
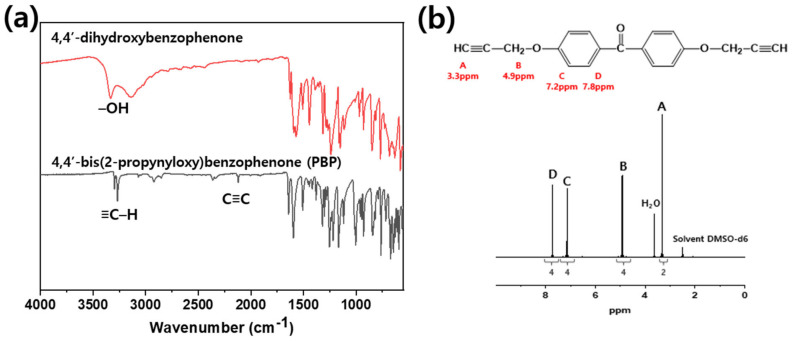
(**a**) FT-IR and (**b**) ^1^H-NMR spectrum of PBP.

**Figure 5 polymers-13-00518-f005:**
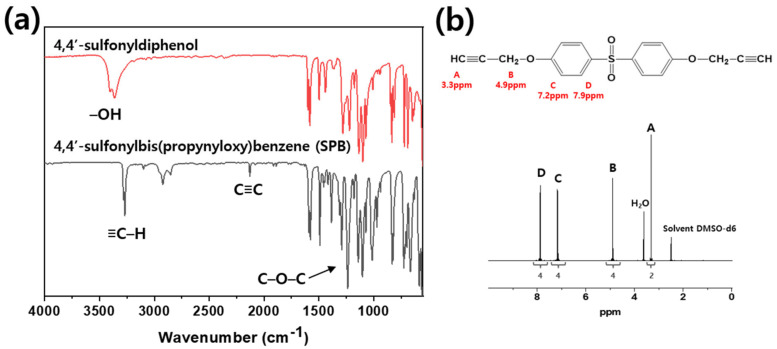
(**a**) FT-IR and (**b**) ^1^H-NMR spectrum of SPB.

**Figure 6 polymers-13-00518-f006:**
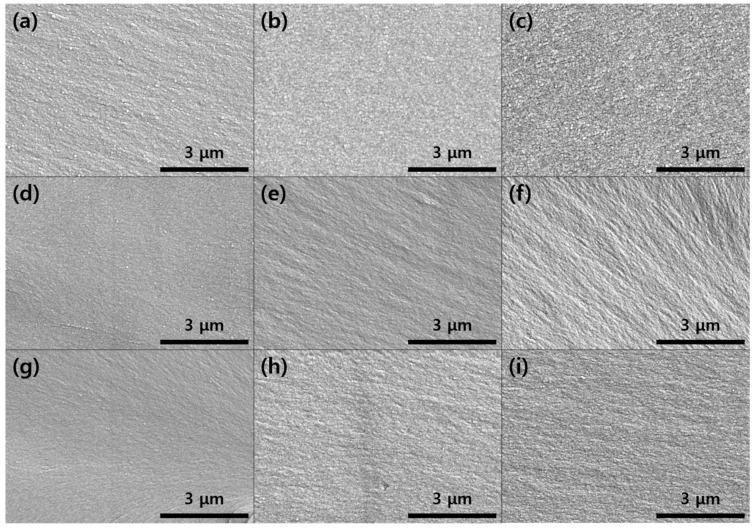
SEM images of the fracture surfaces of the prepared epoxy samples: (**a**) DGEBA; (**b**) DGEBF; (**c**) TGAP; (**d**) DGEBA/PTK; (**e**) DGEBF/PTK; (**f**) TGAP/PTK; (**g**) DGEBA/PTS; (**h**) DGEBF/PTS and (**i**) TGAP/PTS.

**Figure 7 polymers-13-00518-f007:**
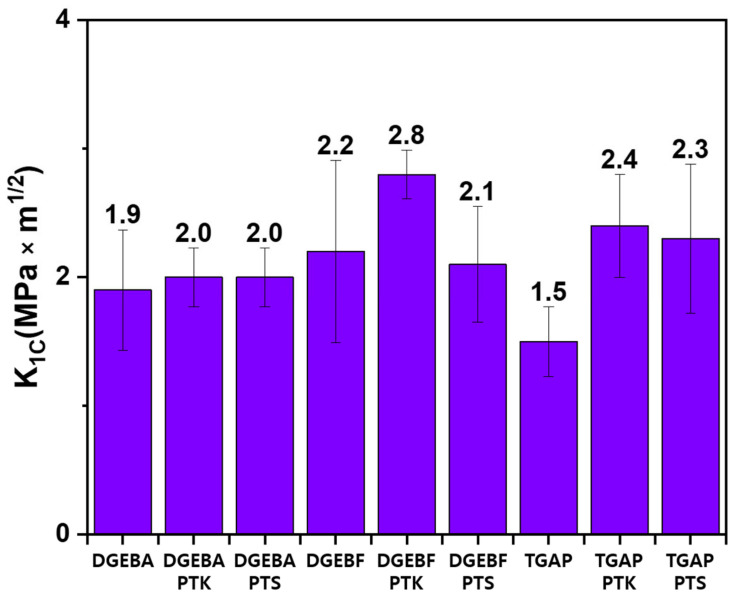
Fracture toughness of the prepared samples.

**Figure 8 polymers-13-00518-f008:**
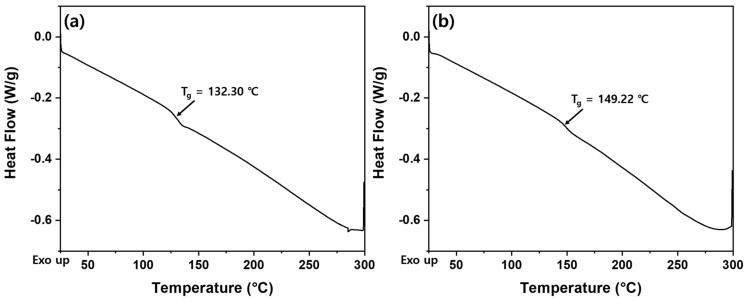
DSC curves of (**a**) PTK and (**b**) PTS.

**Figure 9 polymers-13-00518-f009:**
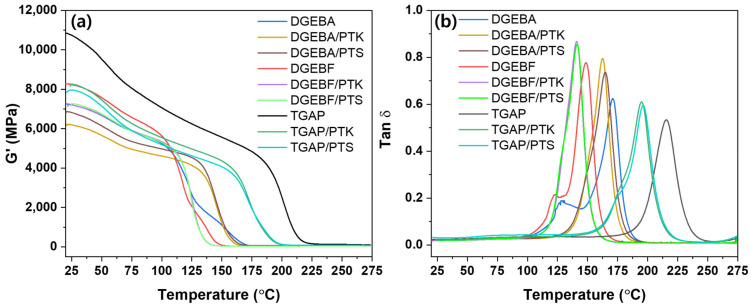
Dynamic mechanical properties of the prepared samples: (**a**) Storage modulus curves; (**b**) tan δ curves.

**Figure 10 polymers-13-00518-f010:**
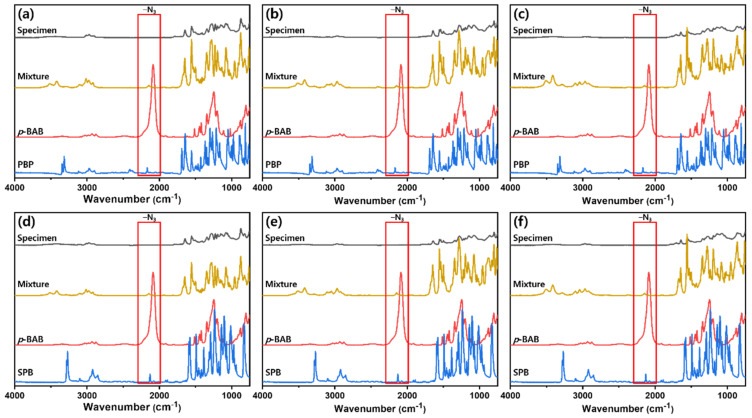
FT-IR spectra of specimen, mixture, p-BAB, and PBP (or SPB): (**a**) DGEBA/PTK; (**b**) DGEBF/PTK; (**c**) TGAP/PTK; (**d**) DGEBA/PTS; (**e**) DGEBF/PTS and (**f**) TGAP/PTS.

**Figure 11 polymers-13-00518-f011:**
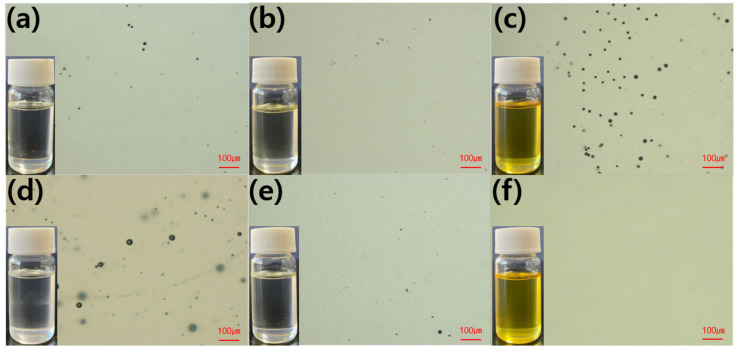
Optical microscope images of the epoxy mixtures: (**a**) DGEBA/PBP; (**b**) DGEBF/PBP; (**c**) TGAP/PBP; (**d**) DGEBA/SPB; (**e**) DGEBF/SPB and (**f**) TGAP/SPB.

**Table 1 polymers-13-00518-t001:** Tensile properties of the prepared samples.

Epoxy Sample	Tensile Strength (MPa)	Tensile Modulus (MPa)
DGEBA	93.5 ± 13.7	1109 ± 173
DGEBA/PTK	89.3 ± 6.4	1032 ± 142
DGEBA/PTS	94.1 ± 4.2	1120 ± 29
DGEBF	86.3 ± 3.4	995 ± 33
DGEBF/PTK	93.7 ± 15.0	1052 ± 157
DGEBF/PTS	100.9 ± 6.0	1107 ± 127
TGAP	116.7 ± 8.2	1259 ± 140
TGAP/PTK	100.9 ± 10.6	1126 ± 129
TGAP/PTS	111.6 ± 11.9	1313 ± 151

**Table 2 polymers-13-00518-t002:** Glass transition temperatures (T_g_s) of the prepared samples.

Epoxy Sample	T_g_ (°C)
DGEBA	171.0
DGEBA/PTK	162.6
DGEBA/PTS	164.6
DGEBF	148.7
DGEBF/PTK	140.9
DGEBF/PTS	141.4
TGAP	215.5
TGAP/PTK	195.1
TGAP/PTS	196.4

## Data Availability

The data presented in this study are available on request from the corresponding author.

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
