# Peer review of "Comparative Study on Toughening Effect of PTS and PTK in Various Epoxy Resins"

_polymers, 2021, doi:10.3390/polym13040518_

Round 1

Reviewer 1 Report

Review of Manuscript ID:Polymers 1094849

Title:Fracture toughness of various epoxy resins toughened with in-situ polytriazoleketone and polytriazolesulfone

In this work,authors present a detailedinvestigationof the difference between the toughening effect of PTK and that of PTS when applied to various epoxy resins, such as diglyc-11 idyl ether of bisphenol A (DGEBA), diglycidyl etherof bisphenol F (DGEBF), and triglycidyl 12 p-aminophenol (TGAP) with 3,3’-diaminodiphenylsulfone as a curing agent.

The manuscript contains some interesting results.However, the authors have recently published very similar articles. Hence, the novelty of this article appears low. Only part of the study such asPTS and PTK inDGEBF resin systemseems to benewin the submitted manuscript. Authors should therefore clearlypoint outthe novelty of their submitted work. Elaboration of old data and resultsshould be avoided as much as possible.

Effect of Polytriazolesulfone Addition on Fracture Toughness of DGEBA Epoxy ResinW Kwon, M Lee, M Han, E Jeong -Textile Coloration and Finishing, Volume 31 Issue 2 / Pages.118-126 / 2019 / 1229-0033(pISSN) / 2234-036X(eISSN)

Effect of Polytriazoleketone-toughening Agent on Mechanical Properties of TGAP/3,3'-DDS Epoxy Resin Textile Science and Engineering Volume 57 Issue 1 / Pages.42-49 / 2020 / 1225-1089(pISSN) / 2288-6419(eISSN)

However, if the editordecidesto proceed with this submission, herearesome initialcomments:

-Can authors providethe investigation of fracture surface of the PES (PTS or PTK) /epoxy resin composites using scanning electron microscopy. It might be useful to see the dispersion state of the PTK or PTSin the epoxy resin.

-Tensile test resultssuch as tensile strain and modulus valuescan beincludeda Table.

-Define -monomer of PTK (PBP) or PTS (SPB) and

-These are used to 50 the monomer forms...would be in

-Improve Fig8so that the transitioncan be clearly seen.

Author Response

Please, check attached file.

Reviewer 2 Report

In my opinion, this is a well-written manuscript that gives a detailed account of an investigation into toughening of epoxy materials.

I have only two, very minor points:

1) L50-51; something is wrong in the sentence 'These are used to the monomer forms in the epoxy resin'.  (Possibly: These are added to the monomer forms in the epoxy resin.)

2) L189: I think 'either' should be 'ether'.

Author Response

Please, check the attached file.

Reviewer 3 Report

The paper is focusing on the toughening of epoxy resins. The decleared goal is to increase the tougheing while not increasing viscosity. The latter properties is not reported at all. The paper has some unclear points that must be addressed before considering for pubblication.

polymers-1094849-

Line 201- this statement is not entirely correct. In the literature there are studies demonstrating that using lower co-PES systems with proper end groups can allow to achieve, by chain extension reactions, proper toughening. See [Cicala, G. et alInfluence of copolymer's end groups and molecular weights on the rheological and thermomechanical properties of blends of novel thermoplastic copolymers and epoxy resins (2006) Journal of Applied Polymer Science, 101 (1), pp. 250-257 ]

Line 211the term “was applied” is not clear better to use “was mixed with”

Fig.6 the fracture toughness improvement should be compared with the common values find for these commercial resins when using standard engineering thermoplastics.

Figure 6 and 7 all the data should be analyzed by using ANOVA to support, statistically, the presence of significant differences.

Lines 232-234. The authors propose an explanation for the toughness increases observed “This result is assumed that the molecular weight of the toughening agent polymerized in the epoxy resin is  sufficiently high to exhibit an excellent toughening effect without decreasing the tensile properties.” The whole phrase is unclear and must be rewritten. In addition, this explanation contrasts somehow with the reported explanation for the toughening behavior observed for PTK in TGAP. In one case some unexplained reaction is supposed to yield the good results observed (..the toughening agent polymerized…) while other cases the molecular weight is ascribed as the cause for the observed toughening behavior. The authors should clarify the mechanism and relevant reaction occurring in their systems.

Fig.9 can be condensed in 2 or 4 pictures only showing directly the effect of the selected blends on the storage modulus or the tan delta (PS I would suggest to use the Greek nomenclature). The presence of minor peaks (fig.9a, 9b) or shoulder (fig  9f, 9i) is not commented at all. The presence of lower peak for pure resins is not commonly observed and must be explained. The presence of shoulder in the DMA of a blend can be a clear sign of phase separation (see Cicala, G.Polymer Engineering and Science Volume 48, Issue 12, December 2008, Pages 2382-2388 Studies on epoxy blends modified with a hyperbranched polyester or Influence of a selected hardener on the phase separation in epoxy/thermoplastic polymer blends; Blanco, I Journal of Applied Polymer Science, 2004, 94(1), pp. 361-371 ) and it must be commented .

Line 270-271 this statement is a relevant par of the paper and should be expanded and improved  with some scheme of the reactions occurring.

Line 303-311. P-BAB , SPB and PBB are used in the synthesis of PTK and PTS. It is not clear if the optical mixtures shown in figure 11 refers to analysis performed on epoxy mixtures with PBP and SPB. Where are the pictures of the resulting blends from mixing of the epoxy with PTK and PTS?

Author Response

Please, check the attached file.

Round 2

Reviewer 3 Report

The paper has been reviewed and the authros improved it acccording to the points raised. The paper can now be considered for pubblication.